

# Centromere protein N may be a novel malignant prognostic biomarker for hepatocellular carcinoma

Qingqing Wang, Xiaoyan Yu, Zhewen Zheng, Fengxia Chen, Ningning Yang and Yunfeng Zhou

Hubei Cancer Clinical Study Center, Hubei Key Laboratory of Tumor Biological Behaviors, Zhongnan Hospital, Wuhan University, Wuhan, China
Department of Radiation Oncology and Medical Oncology, Zhongnan Hospital, Wuhan University, Wuhan, China

Corresponding author
Yunfeng Zhou, yfzhouwhu@163.com

## ABSTRACT

**Background**. Hepatocellular carcinoma (HCC) is one of the deadliest tumors. The majority of HCC is detected in the late stage, and the clinical results for HCC patients are poor. There is an urgent need to discover early diagnostic biomarkers and potential therapeutic targets for HCC.

**Methods**. The GSE87630 and GSE112790 datasets from the Gene Expression Omnibus (GEO) database were downloaded to analyze the differentially expressed genes (DEGs) between HCC and normal tissues. R packages were used for Kyoto Encyclopedia of Genes and Genomes (KEGG) and Gene Ontology (GO) enrichment analyses of the DEGs. A Search Tool for Retrieval of Interacting Genes (STRING) database was used to develop a protein-protein interaction (PPI) network, and also cytoHubba, Molecular Complex Detection (MCODE), EMBL-EBI, CCLE, Gene Expression Profiling Interactive Analysis (GEPIA), and Oncomine analyses were performed to identify hub genes. Gene expression was verified with a third GEO dataset, GSE25097. The Cancer Genome Atlas (TCGA) database was used to explore the correlations between the hub genes and clinical indexes of HCC patients. The functions of the hub genes were enriched by gene set enrichment analysis (GSEA), and the biological significance of the hub genes was explored by real-time polymerase chain reaction (qRT-PCR), western blot, immunofluorescence, CCK-8, colony formation, Transwell and flow cytometry assays with loss-of-function experiments in vitro.

**Results**. Centromere protein N (CENPN) was screened as a hub gene affecting HCC tumorigenesis. Evaluation by Cox regression showed that a high level of CENPN expression was an independent danger variable for poor prognosis of HCC. GSEA showed that high CENPN expression was linked to the following pathways: liver cancer subclass proliferation, cell cycle, p53 signaling pathway, Rb1 pathway, positive regulation of cell cycle G1/S phase transition, and DNA damage response signal transduction by p53 class moderators. Further cell experiments showed that knocking down CENPN expression decreased the proliferation and colony-forming abilities of HepG2 and Huh7 cells as well as Ki67 expression in these cell lines. The cell cycle was arrested in G1 phase, which is consistent with previous experiments on CENPN downregulation., but neither migration nor invasion were significantly affected. Western blot results revealed that the expression of p53, p27, p21, CDK4, cyclin D1, CDK2, cyclin E, pRb, E2F1 and c-myc decreased after CENPN knockdown, but there was no significant change in

total Rb levels. In addition, CENPN-knockdown cells subjected to irradiation showed significantly enhanced of $\gamma$-H2AX expression and reduced colony formation.
**Conclusion**. CENPN functions as an oncogene in HCC and may be a therapeutic target and promising prognostic marker for HCC.

# INTRODUCTION

Liver cancer is among the most common malignant tumors. Despite a declining mortality rate, liver cancer remains to be one of the leading 10 causes of cancer-related fatalities in many countries (*Siegel, Miller & Jemal, 2020*). Hepatocellular carcinoma (HCC) is the most common type of liver cancer, as well as its occurrence and development are closely related to genetic changes, genetic susceptibility and alterations in key signaling pathways (*Marquardt & Thorgeirsson, 2014*). HCC is a highly heterogeneous disease (*Calderaro et al., 2019*); however, due to the limited availability of HCC markers for early diagnosis, most HCC is detected at an advanced disease stage, which limits the effectiveness of common treatment methods such as surgical resection and chemotherapy (*Forner, Reig & Bruix, 2018*). Therefore, it is vital to acquire a better understanding of HCC and also to identify new early diagnostic as well as therapeutic targets.

Increasing knowledge of the diversity and heterogeneity of tumors and the completion of the Human Genome Project have actually promoted the wider use of modern high-throughput sequencing technology. In addition, researchers have begun to use bioinformatics analysis techniques to determine differentially expressed genes (DEGs) and functional paths associated with tumorigenesis (*Can, 2014*).

In this study, by analyzing the GSE87630 and GSE112790 datasets, we found that the DEG CENPN was related to the occurrence of HCC and confirmed this finding in the GSE25097 and also TCGA-LIHC. We observed that upregulated CENPN expression was related to low survival and the advanced T classification of HCC patients. Functional studies showed that knocking down the CENPN gene could inhibit cell proliferation and increase the cytotoxic effect of X-rays on cells in vitro. Further mechanistic studies showed that CENPN partly regulated the cell cycle through the p27/p21-Rb/E2F1 axis.

# MATERIALS & METHODS

## Microarray data

The gene expression profiles of the GSE87630, GSE112790 and GSE25097 datasets were downloaded from the National Center for Biotechnology Information (NCBI) Gene Expression Omnibus (GEO) database (*Edgar, Domrachev & Lash, 2002*). GSE87630 contains 64 HCC samples and 30 normal samples and is based on the GLP6947 platform (Illumina HumanHT−12 V3.0 Expression BeadChip) (*Woo et al., 2017*); GSE112790

contains 183 HCC samples and 15 normal samples and is based upon the GLP570 platform ((HG-U133_Plus_2) Affymetrix Human Genome U133 Plus 2.0 Array) (*Shimada et al., 2019*); and GSE25097 contains 268 HCC samples, 243 adjacent nontumor samples, and 40 cirrhotic liver samples, and also 6 healthy liver samples and is based upon the GPL10687 platform (Rosetta/Merck Human RSTA Affymetrix 1.0 microarray, Custom CDF) (*Tung et al., 2011*).

## Identification of DEGs

DEGs between HCC samples and normal samples were identified utilizing GEO2R (*Barrett et al., 2013*), a tool for the online analysis of differential genes between datasets in a GEO series, with the following thresholds: | logFC | >1 and a false discovery rate (FDR) < 0.05. Volcano plots were constructed with GraphPad Prism 8 (GraphPad Software Inc.). Heatmaps were drawn with the heatmap R package (*Galili et al., 2018*). Overlapping DEGs in the two datasets were detected with the VennDiagram R package (*Chen & Boutros, 2011*).

## Functional enrichment analysis

Gene Ontology (GO) as well as Kyoto Encyclopedia of Genes and Genomes (KEGG) pathway enrichment analyses were mainly utilized to explore the functions of these identified DEGs. GO consists three components: biological process (BP), cellular component (CC) as well as molecular function (MF) (*Ashburner et al., 2000*). KEGG mainly evaluates the pathways in which DEGs may be involved (*Kanehisa & Goto, 2000*). The results were visualized using clusterProfiler and ggplot2 R package (threshold: $P < 0.05$) (*Ito & Murphy, 2013*; *Yu et al., 2012*).

## Protein-protein interaction (PPI) network construction

The PPI network comprising all the DEGs was predicted and analyzed by utilizing the Search Tool for Retrieval of Interacting Genes (STRING) database (http://string-db.org) (*Szklarczyk et al., 2015*) S. A combined score >0.7 was taken into consideration significant. The PPI network was used to clarify mechanisms relevant to the event and growth of HCC.

## Hub gene selection

The public bioinformatics software Cytoscape is a system for visualizing complicated networks and incorporating related information (*Smoot et al., 2011*). The cytoHubba plugin application of Cytoscape was utilized to screen hub genes in a network through 11 topological analysis approaches (*Chin et al., 2014*). In this study, the intersecting sets of the top 30 genes revealed with the maximal clique centrality (MCC) and density of maximum neighborhood component (DMNC) methods were visualized as a Venn diagram (http://bioinformatics.psb.ugent.be/webtools/Venn/). Molecular Complex Detection (MCODE) was utilized to screen within the PPI network with the following parameters: degree cutoff > 2, node score cutoff > 0.2, K-core > 2 and max dep > 100 (*Gary & Bader, 2003*)). Gene Expression Profiling Interactive Analysis (GEPIA) (http://gepia.cancer-pku.cn/index.html), an online tool that consists of information from The Cancer Genome Atlas (TCGA) and Genotype-Tissue Expression (GTEx) databases, was utilized to analyze the expression of the genes as well as their association with patient

survival and clinical stage (*Tang et al., 2017*). Then, after combining the statistical results of gene expression from the Oncomine (http://www.oncomine.com) (*Rhodes et al., 2004*), EMBL-EBI (https://www.ebi.ac.uk) and CCLE (https://www.broadinstitute.org/ccle) databases, the statistically significant hub genes were screened. EMBL-EBI, a web-based tool, provides a series of bioinformatics applications for sequence analysis (*Li et al., 2015*). CCLE contains genomic data, analytical data, and visualization data for approximately 1000 cell lines (*Barretina et al., 2012*). The levels of expression of the hub genes in clinical samples were compared with those in the GSE87630, GSE112790 and GSE25097 datasets using GraphPad Prism 8 (GraphPad Software Inc.). The area under the curve (AUC) assessed by the pROC R package was used to validate the prognostic efficacy of the hub genes in liver cancer (*Robin et al., 2011*).

## Database analysis of the differential expression and prognosis of the CENPN gene

CENPN gene expression in the TCGA liver cancer database (TCGA-LIHC) was extracted. The limma R package was utilized to analyze the different scatter plots, and the survival R package was used to draw survival curves, and also the limma and ggpubr R packages were utilized to analyze CENPN expressing in the different clinical stages (*Holleczek & Brenner, 2013*; *Ritchie et al., 2015*).

## Gene set enrichment analysis (GSEA)

The TCGA-LIHC gene expression dataset was downloaded and installed from the Xena web browser (https://xenabrowser.net/). Two groups from the TCGA-LIHC dataset were classified based on the median CENPN expression level (high and low CENPN). The GSEA 4.1.0 software program, which was downloaded from http://www.broad.mit.edu/gsea/, was utilized to carry out GSEA with the following predefined gene sets (*Subramanian et al., 2005*): hallmark gene sets, curated gene sets and gene ontology. The permutation number was established as 1000, and an FDR <0.25 was considered significant.

## Cell culture and cell transfection

The human HCC cell lines HepG2 and Huh7 were purchased from Procell Life Science & Technology Company (Wuhan, China) and confirmed to contain no mycoplasma contamination via STR analysis. Cells were cultured in DMEM supplemented with 10% fetal bovine serum and 1% penicillin/streptomycin at 37 °C and 5% $CO_2$. A small interfering RNA (siRNA) targeting CENPN and a negative control (NC) were purchased from GenePharma (Suzhou, China). We used Lipofectamine 3000 (L3000015, Invitrogen, Waltham, USA) as the transfection reagent according to the manufacturer's instructions. Forty-eight hours following transfection, the cells were harvested for further experiments, including RNA or protein extraction.

## RNA extraction, reverse transcription as well as qRT-PCR

Total RNA was extracted utilizing the traditional assay, that is to say using TRIzol reagent (Invitrogen, USA). The cDNA synthesis with total RNA (1 µg) was carried out with the Primescript[TM] RT reagent kit (Vazyme, Nanjing, China), and quantitative
real-time PCR (qRT-PCR) was conducted utilizing 2x SYBR$^{®}$ Green Supermix (Vazyme, Nanjing, China). Relative gene expression was quantified via the 2-$^{\Delta\Delta Ct}$ approach and normalized to GAPDH expression. The primer sequences used for amplification were as follows: CENPN forward, 5′-CTGTGTGAGGAAAAGCGTGC-3′; CENPN reverse, 5′-TCACCTGGTCCTTTACTCATCTG-3′; GAPDH forward, 5′-GGAGCGAGATCCCTCCAAAAT-3′, and GAPDH reverse, 5′-GGCTGTTGTCATACTT-CTCATGG-3′.

## X-irradiation

Cells were exposed to X-rays at doses of 0, 2 and 10 Gy according to the experimental design. The equipment used was a Siemens Primus Accelerator (6 Mv; Siemens AG, Munich, Germany)t at Zhongnan Hospital of Wuhan University.

## CCK-8 and colony formation assays

For the CCK-8 assay, cells were seeded into plates with 96 wells at 3000 cells per well. Ten microliters of CCK-8 reagent (Dojindo Laboratories, Kumamoto, Japan) were added to each well, and the plates were incubated at 37 °C for 1–4 h. The absorbance of the wells at 450 nm was determined in a microplate reader (Molecular Devices, USA).

For the colony formation assay, 1000 HepG2 cells per 1 ml in an overall volume of 2 ml was seeded into a plate with six wells. For Huh7 cells, 1,500 cells per 1 ml in a total volume of 2 ml was seeded per well. After 24 h, the cells were irradiated with X-rays at doses of 0 Gy and 2 Gy. After 14 days, the cells were treated with 4% paraformaldehyde for 15 min as well as stained with 0.1% crystal violet for 1 h. Finally, the number of colonies was calculated after washing the wells with water three times and drying.

## Flow cytometry analysis

HCC cells were harvested 48 h following transfection and washed in PBS. The cells were stained with 1 ml of DNA staining solution and 10 µl of propidium iodide (PI) at room temperature for 30 min protected from light and then were analyzed by flow cytometry (Cat. #FC500, Beckman, USA).

## Transwell assay

Cells transfected with siRNA after 24 h were seeded in a Transwell chamber system (Corning, USA). Approximately $2 \times 10^5$ HepG2 cells or $8 \times 10^4$ Huh7 cells in 200 µl of serum-free medium were seeded in the upper chamber, while 600 µl of medium containing 20% serum was added to the lower chamber. After a 48-hour incubation, cells in the upper chamber were wiped off with cotton swabs, and those in the lower chamber were fixed and stained as described in the colony formation experiment before they were photographed under an optical microscope.

## Western blotting

Protein lysates were obtained from HepG2 and Huh7 cells with RIPA lysis buffer (#P0013B, Beyotime Biotechnology, China) containing 1% cocktail (#HY-K0021, MCE, USA). The protein concentration was detected with a BCA assay kit (Beyotime Biotechnology, Shanghai, China), and 30 µg of protein sample was divided by SDS-PAGE through a 10%

gel and then transferred to PVDF membranes, which were blocked in 5% skim milk in TBST before they were blotted with the appropriate primary antibody followed by the corresponding secondary antibody. Detailed information on the antibodies used is listed in Table S1. An enhanced chemiluminescence kit was utilized to develop the bands.

## Immunofluorescence staining

The immunofluorescence method was based on a previously described method (*Wang et al., 2017*). Briefly, cells were seeded into a 6-well plate or on cell slides and cultured overnight. After fixation for 15 min, the cells were incubated with 0.2% Triton X-100 for 15 min at RT and blocked with 5% BSA for 30 min. Next, the cells were incubated successively with primary antibody targeting CENPN, E-cadherin, N-cadherin, vimentin or $\gamma$-H2AX and appropriate secondary antibody before they were stained with DAPI at RT for 5 min. Finally, the cells were photographed under a fluorescence or confocal microscope.

## Statistical analysis

All prognostic information from the TCGA-LIHC dataset was collected, and samples without results were excluded. On the basis of optimal sample separation, Kaplan–Meier (K-M) survival curves were generated to calculate survival, and the difference in survival between groups was determined by the log-rank test. The relationships between variables and patient survival were analyzed with Cox models, and the results were evaluated with SPSS 22.0 software (SPSS, Chicago, IL, United States). All cell experimental results were individually replicated 3 times. The values are reported as the means $\pm$ standard deviations and were statistically analyzed with GraphPad Prism 8.0 software (GraphPad Software, La Jolla, CA, United States). Student's $t$-test was utilized for comparisons between groups. $P < 0.05$ was thought about statistically significant.

# RESULTS

## Preliminary screening and enrichment analysis of DEGs in HCC

Figure 1 illustrates the study design. The DEGs in HCC vs. normal tissues in the GSE87630 and GSE112790 datasets were analyzed by processing and standardizing the gene expression profile data through GEO2R (Figs. 2A, 2B, Figs. S1 and S2). GSE87630 contains 1,162 DEGs, of which 394 are upregulated and 768 are downregulated, whereas GSE112790 contains 1,713 DEGs comprising 963 upregulated DEGs and 750 downregulated DEGs. Then, R was used to intersect the two datasets, and a total of 532 overlapping DEGs were acquired: 171 upregulated genes and 361 downregulated genes (Fig. 2C, Table S2).

To clarify the functions of these 532 DEGs, GO as well as KEGG enrichment were carried out, and the ggplot2 package of R was used to visualize the top 10 per classification of GO and the top 35 of KEGG. In the GO BP category, the identified DEGs were related to the organic acid catabolic process and epoxygenase P450 pathway (Fig. 2D, Table S3). In the GO CC category, the DEGs were enriched in MCM complex and blood microparticle extracellular matrix (Fig. 2D, Table S3). Finally, in the GO MF classification, the DEGs were enriched in monooxygenase activity and iron ion binding (Fig. 2D, Table S3). KEGG pathway enrichment result showed that these DEGs play important roles in multiple key

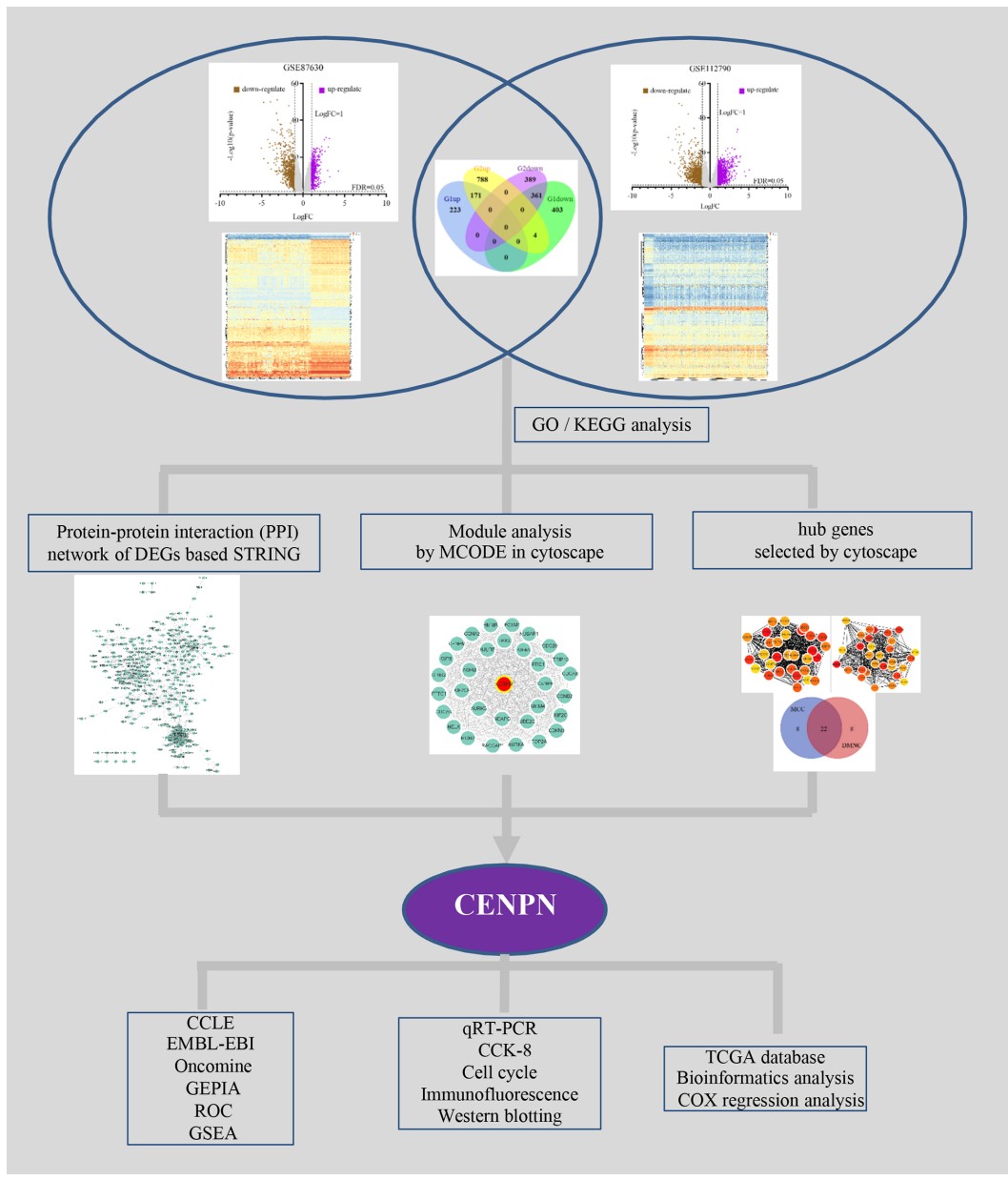

**Figure 1** Flowchart of the integrated analysis.

signaling pathways, including DNA replication, the cell cycle as well as the p53 signaling pathway (Fig. 2E, Table S4).

## Construction of the PPI network as well as screening of hub genes

The STRING database was utilized to analyze the potential interacting proteins among 532 DEGs with a combined score > 0.7. For the PPI network, which contained 347 nodes and 1560 edges, was built with Cytoscape software (Fig. 3A). Then, the hub genes were analyzed with the cytoHubba plugin. Two topological analysis approaches, MCC and DMNC, were
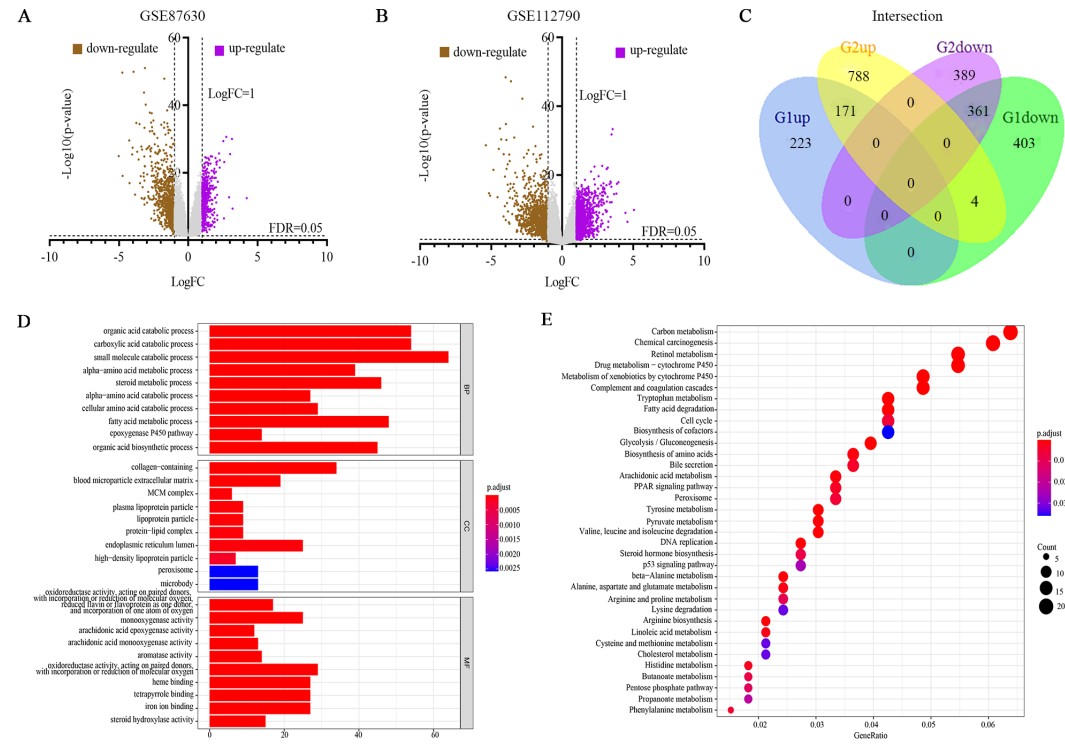

**Figure 2** **Differentially expressed genes (DEGs) were evaluated by Gene Ontology (GO) and Kyoto Encyclopedia of Genes and Genomes (KEGG) analyses.** (A, B) Volcano plot of the DEGs between HCC and normal liver tissues in each dataset. Purple dots: significantly upregulated genes in HCC; brown dots: significantly downregulated genes in HCC; gray dots: non-DEGs. Genes with an adjusted $P$ value (FDR) < 0.05 and |logFC| > 1 were considered to have statistically significant differences in expression. (C) Venn diagram of 532 overlapping DEGs from G1 (GSE87630) and G2 (GSE112790): 171 upregulated genes and 361 downregulated genes. (D) GO analysis. (E) KEGG pathway analysis.

used to rank the top 30 nodes in the created PPI network (Figs. 3C and 3D, Table 1). Then, the intersection was calculated online using a Venn diagram (Fig. 3E, Table S5). Module analysis was performed with MCODE, and the top 5 modules were listed (Table S6). CENPN was screened as the seed gene in module 1 (the most significant module), which consisted of 32 nodes as well as 479 edges (Fig. 3B, Table S7). Thus, CENPN was considered the hub gene.

## Study of CENPN expression in HCC cells and tissues via online databases

CCLE and EMBL-EBI were used to explore the RNA expression level of CENPN in HCC cell lines. The results revealed that CENPN was highly expressed in liver cancer cells (Figs. 4A and 4B). With respect to tissue expression levels, Oncomine evaluation revealed that CENPN expression was significantly higher in tumor tissues than in normal tissues (Fig. 4F), and CENPN expression in the GSE87630 and GSE112790 datasets was consistent with the above results (Figs. 4C and 4D); furthermore, this result was verified in the GSE25097 dataset (Fig. 4E). GEPIA revealed that the overall survival (OS) and disease-free
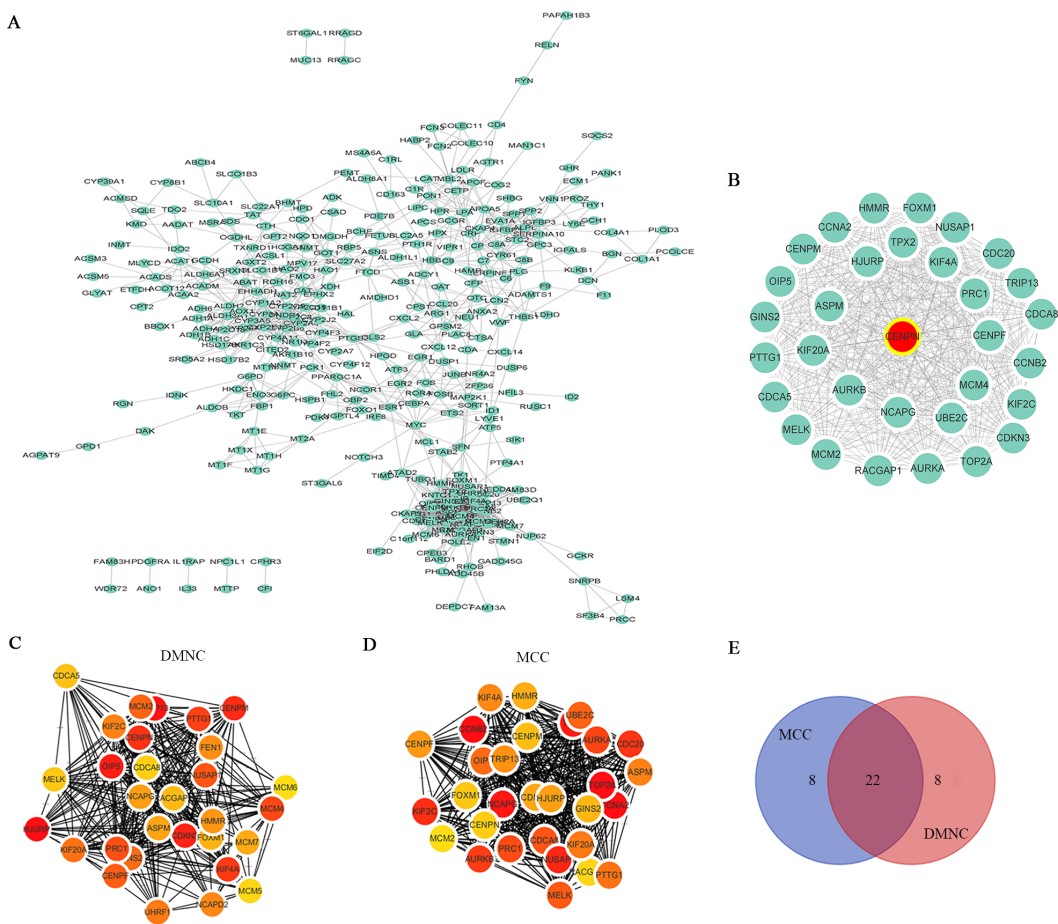

**Figure 3** **Screening the hub gene through STRING and Cytoscape.** (A) A total of 347 DEGs were visualized in a PPI network. Nodes represent proteins, and edges represent interactions among proteins. There were 347 nodes and 1560 edges in the network. (B) The most significant module in MCODE analysis. (C, D) The top 30 hub genes were searched through two ranking methods in cytoHubba. MCC, maximal clique centrality; DMNC, density of maximum neighborhood component. (E) Venn diagram of 22 overlapping hub genes from the MCC and DMNC analyses.

survival (DFS) of liver cancer patients with high CENPN expression were shorter than those of patients with low CENPN expression (Figs. 4G and 4H). To better understand the accuracy of CENPN in HCC tumorigenesis, receiver operating characteristic (ROC) curves were drawn, and the AUC values of GSE87630, GSE112790 and GSE25097 were 0.913, 0.904 and 0.787, respectively (Figs. 4I–4K). Our results revealed that CENPN was highly expressed in liver cancer as well as affected the prognosis of patients.

## Verification of the correlation between CENPN and clinicopathological features of HCC patients through the TCGA database

The optimal cutoff value was calculated by the survminer package of R, and the patients were separated into 2 groups: high CENPN expression and low CENPN expression. Compared with the low expression group, the high expression group had dramatically

**Table 1   Hub genes for DEGs ranked in cytoHubba plugin of Cytoscape.**

| Catelogy | Rank methods in cytoHubba | |
|---|---|---|
| | MCC | DMNC |
| Gene symbol top 30 | *CCNA2* | *CDKN3* |
| | *CDKN3* | *KIF20A* |
| | *KIF20A* | *PRC1* |
| | *PRC1* | *RACGAP1* |
| | *RACGAP1* | *KIF2C* |
| | *KIF2C* | *CDCA5* |
| | *UBE2C* | *CDCA8* |
| | *CDCA8* | *ASPM* |
| | *CDC20* | *HJURP* |
| | *TPX2* | *OIP5* |
| | *AURKA* | *MCM7* |
| | *ASPM* | *MCM6* |
| | *HJURP* | *GINS2* |
| | *TOP2A* | *CENPN* |
| | *OIP5* | *CENPM* |
| | *GINS2* | *MCM4* |
| | *CENPN* | *HMMR* |
| | *CENPM* | *NCAPD2* |
| | *HMMR* | *NCAPG* |
| | *NCAPG* | *NUSAP1* |
| | *AURKB* | *MELK* |
| | *NUSAP1* | *TRIP13* |
| | *MELK* | *FOXM1* |
| | *CCNB2* | *MCM2* |
| | *TRIP13* | *MCM5* |
| | *FOXM1* | *KIF4A* |
| | *MCM2* | *FEN1* |
| | *KIF4A* | *PTTG1* |
| | *PTTG1* | *CENPF* |
| | *CENPF* | *UHRF1* |

**Notes.**
Bold gene symbols were the overlap hub genes in top 30 by two ranked methods respectively in cytoHubba. MCC Maximal clique centrality, DMNC Density of Maximum Neighborhood Component.

shorter survival (Fig. 5C). When the clinical features of CENPN were combined in the analysis, the Wilcoxon rank-sum test revealed that CENPN expression in tumor tissues was dramatically higher than that in normal (Fig. 5A). Similar results were obtained in the paired analysis of normal and tumor tissues from the same patient (Fig. 5B). These results suggest that CENPN expression is negatively associated with HCC prognosis. In particular, with the progression of grade, stage and T stage, CENPN expression showed an upward trend (Figs. 5D–5F). Furthermore, univariate Cox analysis revealed that CENPN, stage, and T and M stages affected HCC prognosis (Figs. 6A–6C, Table 2). Multivariate Cox analysis showed that high CENPN expression was an independent predictor of poor

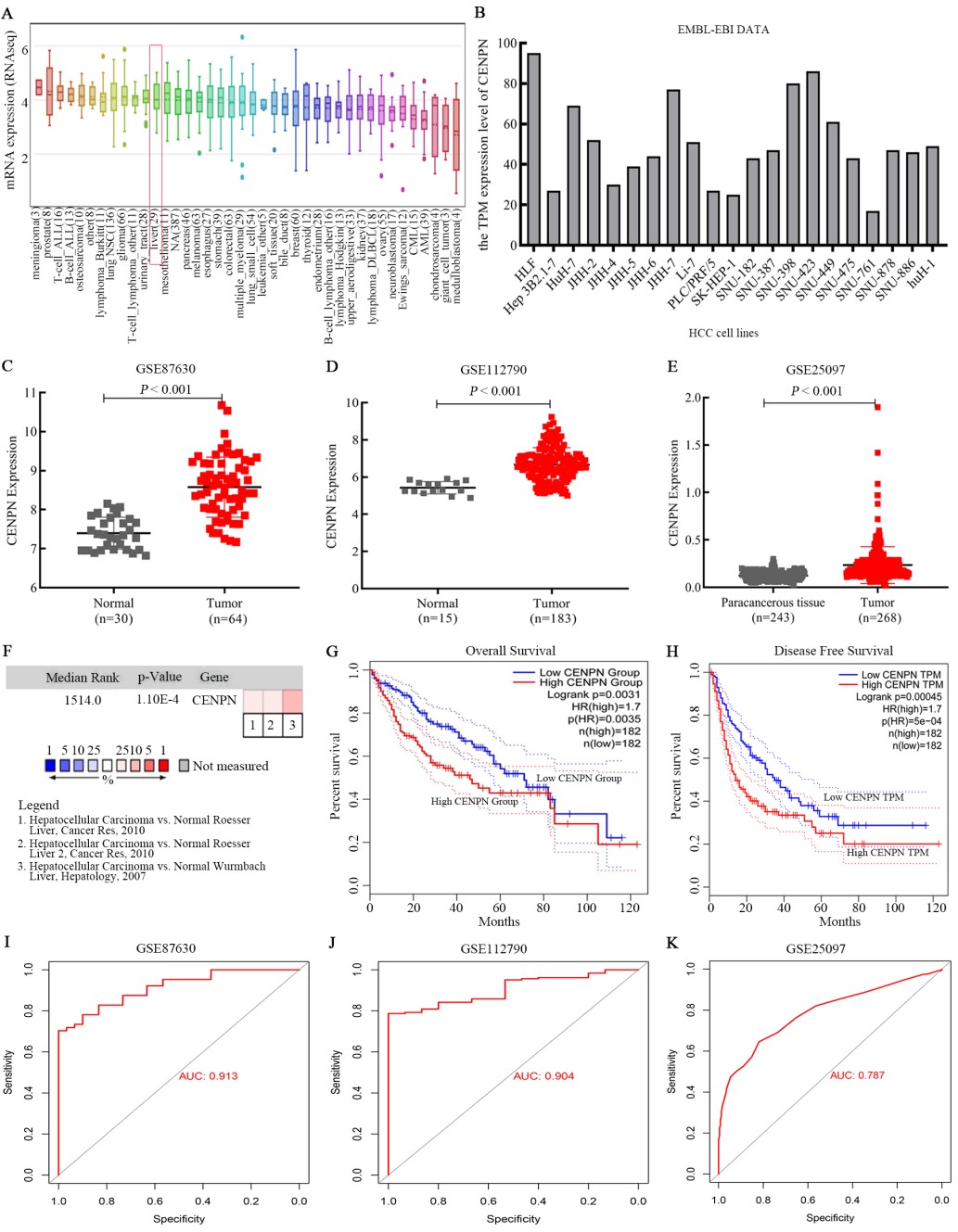

Figure 4  CENPN expression in liver hepatocellular carcinoma and its prognostic information. (A)
CCLE revealed that CENPN was highly expressed in many cancer cells, including liver cancer cells. (B)
The EMBL-EBI results, illustrated with GraphPad Prism 8, showed that CENPN was highly expressed
in many HCC cell lines. (C, D and E) CENPN expression in HCC and liver tissues in the GSE87630,
GSE112790 and GSE25097 datasets. (F) Analysis of CENPN in cancer vs. normal tissue from Oncomine.
Heat maps of hub gene expression in clinical HCC tissues vs. normal liver tissues are shown. (G, H) The
GEPIA database was used to examine the relationship between CENPN expression and overall survival
and disease-free survival in LIHC patients. LIHC, liver hepatocellular carcinoma. HCC, hepatocellular
carcinoma. (I, J and K) The ROC curves of GSE87630, GSE112790 and GSE25097. The AUCs were 0.913,
0.904 and 0.787, respectively. ROC, receiver operating characteristic. AUC, area under the curve.

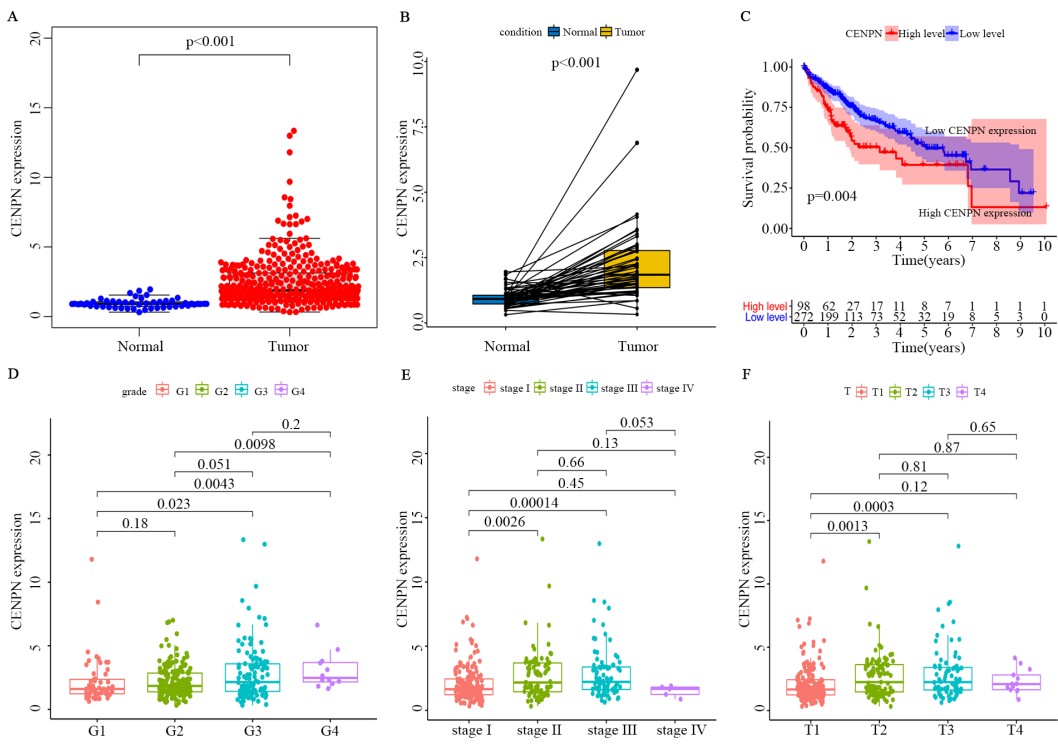

**Figure 5** **Verification of CENPN expression in HCC patients and its correlation with survival and clinicopathological stage using the TCGA database.** (A) The differential expression of CENPN in normal and tumor tissues. All normal and tumor samples were analyzed by the Wilcoxon rank-sum test ($p < 0.001$). (B) Paired differentiation analysis of CENPN expression in normal and tumor samples derived from the same patient ($p < 0.001$ by the Wilcoxon rank-sum test). (C) Survival analysis of LIHC patients with different CENPN expression levels. Patients were classified as having high or low expression according to the optimal cutoff, 3.006547, which was calculated with the survminer package. $P = 0.004$ by the log-rank test. (D–F) Relationship between CENPN expression and clinicopathological stage. The Wilcoxon rank-sum test or the Kruskal–Wallis rank-sum test was used for statistical processing.

prognosis in HCC patients (Table 2). In summary, CENPN acts as an oncogene in liver cancer.

## GSEA of the CENPN gene

GSEA showed that high CENPN expression was mainly associated with liver cancer subclass proliferation, the cell cycle, the p53 signaling pathway, the Rb1 pathway, cell cycle checkpoints, positive regulation of the G1/S phase transition, reactome SCF SKP2-mediated degradation of p27 p21, E2F targets and more correlated gene sets ($P < 0.05$; Figs. 7A–7H). High CENPN expression was also associated with DNA damage, DNA damage detection, and DNA damage response signal transduction by p53 class mediators and to other DNA damage and repair-related gene sets ($P < 0.05$; Fig. 8A).

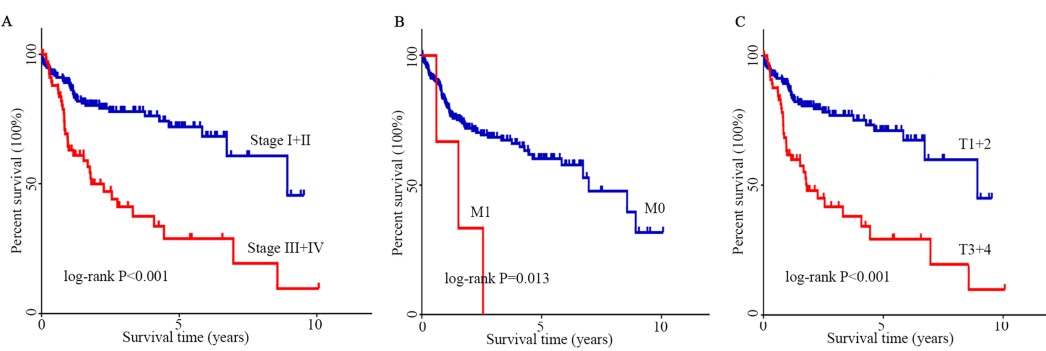

**Figure 6 K–M analysis of clinicopathological parameters.** (A–C) The effect of stage, M stage and T stage on survival according to the K–M curve. The log-rank test was used for statistical processing. Abbreviations: K-M, Kaplan–Meier; M, migration; T, tumor size.

**Table 2 Cox regression analysis of overall survival of HCC.**

| Variables | Univariate analysis | | | Multivariate analysis | | |
|---|---|---|---|---|---|---|
| | HR | 95% CI of HR | *P*-value | HR | 95% CI of HR | *P*-value |
| Age ( ≤65 vs >65) | 0.989 | 0.606–1.614 | 0.964 | 0.875 | 0.521–1.470 | 0.614 |
| Gender (female vs male) | 1.285 | 0.804–2.054 | 0.294 | 1.141 | 0.691–1.883 | 0.606 |
| Grade (grade 1 + 2 vs grade 3 + 4) | 0.934 | 0.592–1.474 | 0.769 | 0.839 | 0.524–1.344 | 0.466 |
| Stage (stageI + II vs stageIII + IV) | 0.324 | 0.206–0.512 | **0.000** | 3.568 | 0.190–66.959 | 0.395 |
| T (T1 + T2 vs T3 + T4) | 0.322 | 0.204–0.509 | **0.000** | 0.098 | 0.005–1.772 | 0.116 |
| M (M0 vs M1) | 0.255 | 0.080–0.813 | **0.021** | 0.411 | 0.118–1.435 | 0.163 |
| N (N1 + N2 vs N3 + N4) | 0.484 | 0.118–1.982 | 0.313 | 0.213 | 0.028–1.634 | 0.137 |
| CENPN expression (low vs high) | 0.593 | 0.368–0.955 | **0.032** | 0.608 | 0.372–0.995 | **0.048** |

**Note.**
Bold indicates statistical significance, $P < 0.05$.

## CENPN downregulation arrests the cell cycle in G1 phase and inhibits HCC cell proliferation but has no effect on migration and invasion

The above results suggest that CENPN functions as a biomarker for the diagnosis of HCC. To further explore the mechanism by which CENPN impacts the biological behavior of HCC, we established a cell model of CENPN deficiency (HepG2 and Huh7) by transfecting siRNA targeting CENPN and NC siRNA. qRT-PCR, western blot and immunofluorescence analyses were performed 48 h after transfection to confirm knockdown (Figs. 7I, 7J, 7M and 7O). CCK-8 assays revealed that CENPN deficiency considerably inhibited the proliferation and viability of HCC cells (Figs. 7K, 7L), and immunofluorescence analysis showed that when CENPN was knocked down, Ki67 expression decreased significantly (Figs. 7N, 7P). Cell cycle analysis showed that CENPN deficiency significantly inhibited the transition from G1 to S phase in HCC cells (Figs. 7Q–7X). However, Transwell assays revealed that there was no difference in the numbers of cells that migrated or invaded after CENPN knockdown compared with those subjected to NC siRNA (Figs. S3A, S3B). Immunofluorescence results indicated that there was no significant difference in E-cadherin, N-cadherin and vimentin expression between the groups (Figs. S3C–S3G), and the western blot results

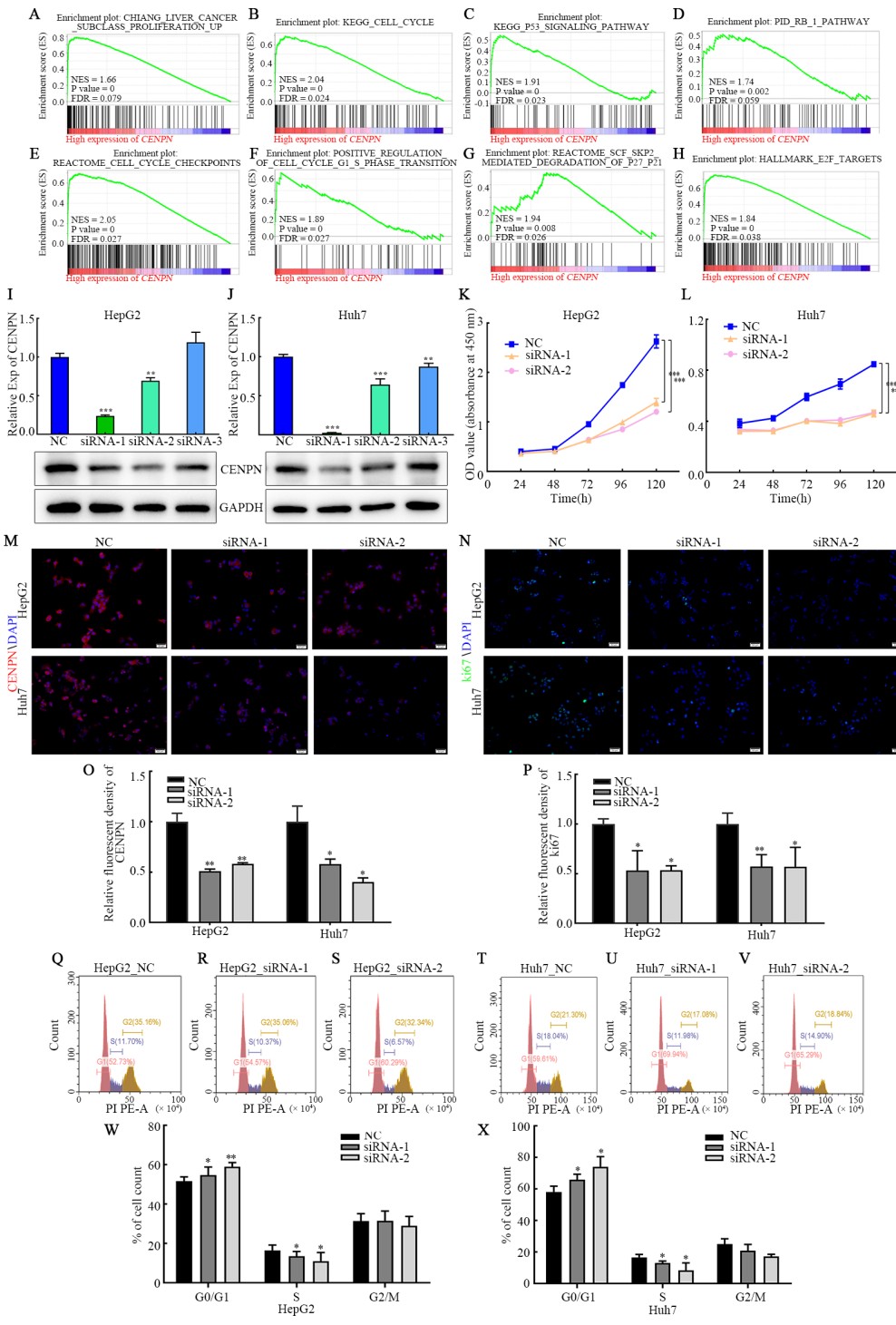

**Figure 7 Interfering with CENPN expression inhibits the proliferation of HCC cells and the G1/S transition.** (A–H) GSEA of CENPN in HCC. (I, J) qRT-PCR and western blot assays of the interference efficiency of CENPN. (K, L) CCK-8 assay of the proliferation ability of cells with CENPN knockdown. (M, O) Immunofluorescence was used to detect the efficiency of siRNA targeting CENPN. (N, P) Immunofluorescence was used to detect the fluorescence intensity of Ki67 in cells with CENPN knockdown. (Q-X) Flow cytometry was used to detect changes in the cell cycle distribution after CENPN knockdown. Each data point represents the mean ± SD from three independent experiments. * $p < 0.05$. ** $p < 0.001$. *** $p < 0.0001$. Scale bars, 50 μm.

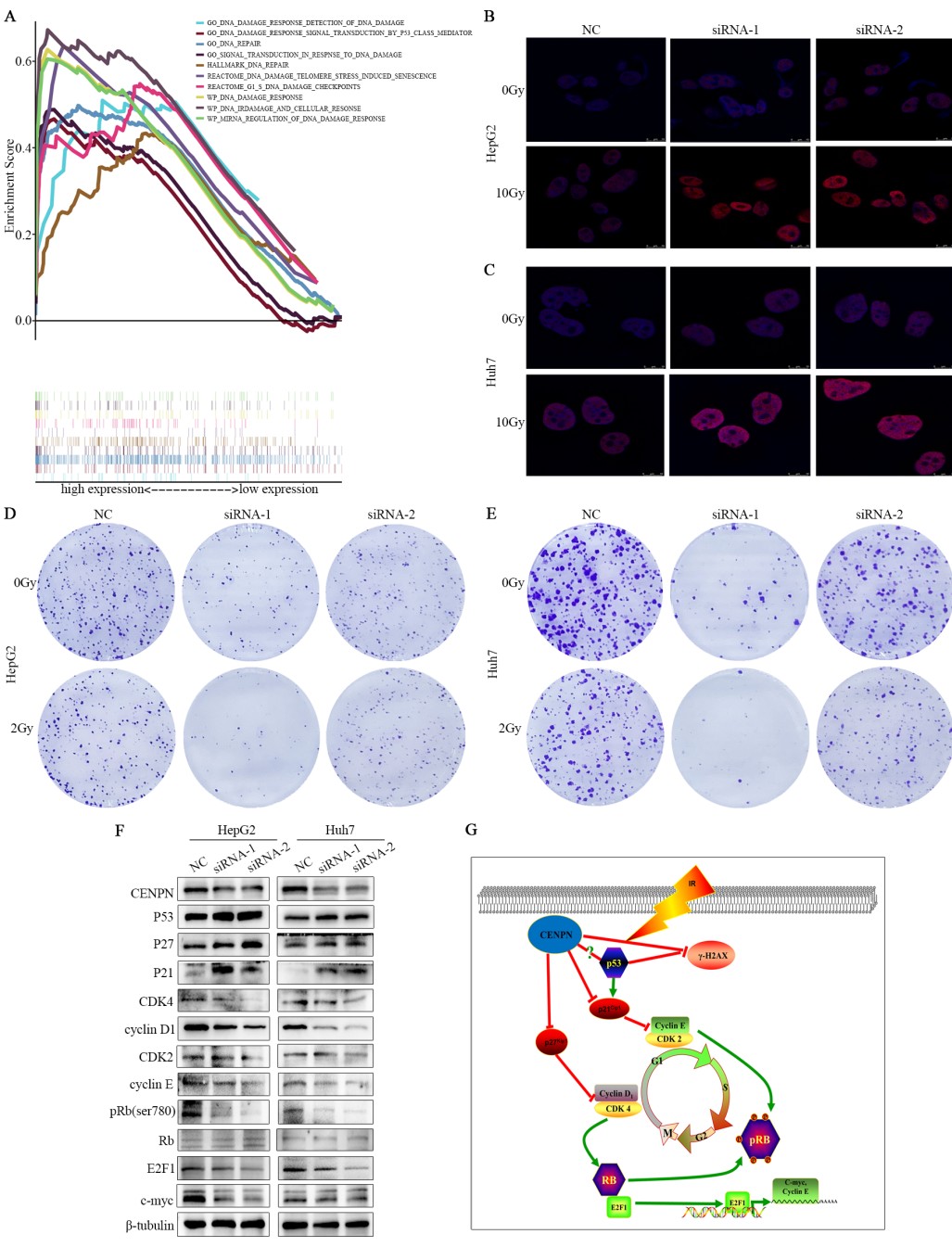

**Figure 8  Exploring the mechanism of CENPN in HCC cells.** (A) GSEA of CENPN in HCC. (B, C) Immunofluorescence detection of $\gamma$-H2AX foci. (D, E) Colony formation assays were used to detect the number of clones in HCC cells after CENPN knockdown alone or combined with 2 Gy X-rays. (F) Western blot assays were performed to detect the expression of G1/S phase-related checkpoint proteins and corresponding upstream and downstream markers in cells with CENPN knockdown. (G) Mechanistic diagram of the biological function of CENPN in HCC cells. Each data point represents three independent experiments. Scale bars, 10 $\mu$m.

were consistent with the above data (Figs. S3H, SI). In summary, these findings indicate that CENPN knockdown inhibits HCC cell proliferation but does not affect invasion or migration.

### CENPN affects the p21-CDK2/cyclin E, p27-CDK4/cyclin D and Rb/E2F1 signaling pathways in HCC

To elucidate the mechanism by which CENPN promotes HCC, we used western blotting to analyze the expression levels of related markers. The western blot results revealed that compared with the NC, the CENPN-knockdown group showed enhanced expression of P53, P27 and P21 and also lowered expression of CDK4, cyclin D1, CDK2, cyclin E as well as pRb, but there was no significant change in total Rb levels. In addition, the expression of E2F1 and c-Myc decreased (Fig. 8F). Taken together, these results indicate that CENPN downregulation prevents cell proliferation via regulating p21 and p27.

### Decreased CENPN expression promotes radiation damage in HCC cells

GSEA indicated that CENPN is closely related to DNA damage and repair functions. Cells transfected with siRNA-CENPN and NC for 48 h were treated with X-rays at doses of 0 Gy and 10 Gy, and $\gamma$-H2AX foci were detected by immunofluorescence 24 h later. The number of $\gamma$-H2AX foci increased after CENPN knockdown and was further pronounced upon 10 Gy X-ray exposure (Figs. 8B, 8C). Moreover, colony formation assays suggested that the number of HCC cell colonies formed after CENPN knockdown combined with X-ray treatment was significantly lower than that after siRNA-CENPN treatment alone (Figs. 8D, 7E). These results suggest that interfering with CENPN expression enhances X-ray-induced radiation damage in HCC cells.

## DISCUSSION

The prognosis of HCC patients is poor, and a major of patients can only receive palliative treatment. The effect of surgical treatment is satisfactory, but few patients benefit from it. Only 1/3 of patients are eligible for radical treatments such as percutaneous ablation, surgical resection or liver transplantation (*Forner, Llovet & Bruix, 2012*). In addition, patients often miss the optimal timeframe for surgery when they are diagnosed, so there is an urgent need to identify new biomarkers and develop new treatment strategies.

In this study, through a comprehensive bioinformatics analysis, 532 DEGs were identified in the GSE87630 and GSE112790 datasets. GO and KEGG analyses showed that the DEGs were closely related to organic acid catabolic process, epoxygenase P450 pathway, MCM complex, blood microparticle extracellular matrix, DNA replication, the cell cycle, the p53 signaling pathway and additional items. PPI network and Cytoscape analyses revealed CENPN as a new DEG in HCC.

CENPN is located on chromosome 16q23.2 and encodes nucleosome-related complexes, which are critical for dynamic assembly. CENPN can recognize the histone H3 variant CENPA (a centromere-specific nucleosome that plays a fundamental role in centromere assembly) in the centromeric nucleosome, and it is correctly controlled by the formation of the CENPA nucleosome-associated complex (NAC) to regulate cell mitosis (*Chittori et al.,*

*2018*; *Foltz et al., 2006*). This correct assembly ensures accurate chromosome separation and avoids diseases caused by aneuploidy due to chromosome misseparation (*Kops, Weaver & Cleveland, 2005*). The consumption of CENPN can lead to the downregulation of several CENPs, and this downregulation is considered a necessary condition for the manufacture of new centromeres (*Tian et al., 2018*). CENPN is closely related to the occurrence and development of many kinds of cancer. Sarah An et al. showed by multivariate analysis that an increase in CENPN expression could significantly increase the recurrence and mortality rates of breast cancer patients with a smoking history (*Andres et al., 2015*). Moreover, *Wright et al. (2017)* identified CENPN as a gene associated with aneuploidy, genomic instability and cancer susceptibility based on intensity data and sequences of genotypic arrays. CENPN regulates cell proliferation and cell cycle progression by affecting the mitotic process (*Wright et al., 2017*). *Rahman et al. (2019)* identified CENPN as a poor prognostic marker for colorectal cancer by bioinformatics analysis. Recently, it was reported that CENPN can promote the proliferation of oral cancer cells in the entrance cavity by regulating the cell cycle (*Oka et al., 2019*). However, the expression of CENPN in HCC and its role and mechanism in malignant development are unclear.

As verified by a third GEO dataset and TCGA data, CENPN is highly expressed in HCC, and the expression level of CENPN was positively associated with grade, stage and T classification and negatively correlated with patient prognosis. Further Cox regression analysis showed that CENPN is a potential independent prognostic variable for HCC. In summary, CENPN may be an oncogene in HCC.

To better understand the function of CENPN in HCC, we used GSEA and found that in terms of the hallmarks of HCC, CENPN was related to the cell cycle, DNA damage and repair, and also other functional items. Moreover, the findings revealed that abnormal expression of the CENPN gene is related to the incident and also progression of HCC. This research is the first to report the promoting effect of CENPN on the growth of human HCC cells. CENPN downregulation can hinder the proliferation of HCC cells and cause G0/G1 phase arrest. The results of our cell experiments suggest that CENPN plays a role as an oncogene. In addition, we examined the expression of epithelial-mesenchymal transition (EMT)-related markers (E-cadherin, N-cadherin as well as vimentin) to understand the oncogenic function of CENPN as fully as possible (*Hugo et al., 2007*); however, there were no difference in expression.

To much better understand the potential molecular mechanism of CENPN in HepG2 and Huh7 cells, we focused on three signaling pathways according to the GSEA results—P27, P21 and Rb/E2F1, which play a vital role in HCC and various cancers (*Gartel, 2009*; *Razavipour, Harikumar & Slingerland, 2020*; *Rubin, Sage & Skotheim, 2020*). Our results show that interfering with CENPN expression in HCC can inhibit c-myc and cyclin E expression by activating the p27-CDK4/cyclin D1 and p21-CDK2/cyclin E axes, thus reducing the level of phosphorylated Rb and inhibiting E2F1 transcription. P21 (*Eldeiry et al., 1993*; *Harper et al., 1993*) and p27 (*Polyak et al., 1994*; *Toyoshima & Hunter, 1994*) are well-known cyclin-dependent kinase inhibitors that mediate cell cycle arrest (*Nan, Jing & Gong, 2004*). When p21 and p27 interact with their respective cyclin binding companions, they block the kinase activity of CDKs (*Abukhdeir & Park, 2008*). P21 prevents the binding

of cyclin E and CDK2 (*Gu, Turck & Morgan, 1993*) and is directly regulated by p53 (*Abukhdeir & Park, 2008*) while p27 prevents the binding of cyclin D1 and CDK4 (*Hunter & Pines, 1994*); therefore, p21 and p27 inhibit the G1/S phase transition (*Alt, Gladden & Diehl, 2002*; *Cheng et al., 1999*). The inactivation of these complexes results in other proteins not being phosphorylated, including pRb (RB1), so it is unable to release E2F transcription factors to activate the expression of genes that regulate S phase (*Matsuoka et al., 1994*). Previous research has revealed that the negative regulation of cyclin D1 and CDK4 is consistent with the downregulation of CENPN (*Molenaar et al., 2008*), and recent reports in oral squamous cell carcinoma (OSCC) suggest that downregulation of CENPN can reduce the expression of cyclin D1 and CDK4 with corresponding increases in the expression of p21 and p27 (*Oka et al., 2019*). This suggests that interfering with CENPN expression in OSCC can upregulate p27 and p21 expression to inhibit CDK4 and cyclin D1 expression and arrest the cell cycle in G1 phase. In addition, some researches have revealed that decreased p27 and p21 expression is closely correlated with the histological grade, metastasis and prognosis of HCC (*Fiorentino et al., 2000*; *Qin & Ng, 2001*). In papillary carcinoma (PTC), CITED1-meditaed interference of p21 and p27 expression can increase the level of phosphorylated Rb and the transcriptional activity of E2F1, which leads to PTC cell proliferation (*Li et al., 2018*). *Xing et al. (2020)* reported that in HCC cells, haprolid upregulates the expression of p21 and p27 and inhibits the G1/S phase transition of cells, which may be related to the downregulation of Rb/E2F expression. Therefore, it can be speculated that CENPN reduces the level of phosphorylated Rb and inhibits E2F1 transcription by downregulating the activities of the p27-CDK4/cyclin D1 and p21-CDK2/cyclin E axes, thus inhibiting c-myc as well as cyclin E, inhibiting cell proliferation, and ultimately affecting the prognosis of liver cancer (Fig. 8G). However, the potential mechanism by which CENPN inactivates p27 and p21 and whether it regulates p21 by inhibiting p53 require further study.

More importantly, we found that knocking down CENPN expression increased X-ray-induced $\gamma$-H2AX expression. $\gamma$-H2AX is an important indicator of radiation-induced DNA double-strand breaks (DSBs) (*Hohmann et al., 2018*). Disruptions in the p53 protein are associated with greater than 50% of human cancers and, p53 is a crucial molecule involved in the cellular response to radiotherapy (*Cuddihy & Bristow, 2004*; *Levine & Oren, 2009*). GSEA suggested that CENPN could be enriched in the p53 pathway, and western blot analysis showed that p53 expression increased with CENPN knockdown, so we inferred that CENPN interferes with DNA damage by inhibiting p53 in HCC (Fig. 8G).

## CONCLUSION

In conclusion, our work showed for the first time that CENPN plays a carcinogenic role in HCC tumorigenesis. The upregulation of CENPN expression was significantly associated with poor patient survival, and knocking down CENPN expression inhibits cell proliferation, at least in part through the p27-CDK4/cyclin D1, p21-CDK2/cyclin E and Rb/E2F1 pathways. In addition, CENPN knockout increases radiotherapy-induced DNA damage. Therefore, CENPN can be used as not only a useful biomarker for diagnosis and

prognosis but also a potential therapeutic target for HCC. In future work, we will extend to the regulatory network of CENPN, such as miRNA and transcription factors involved in regulation. Moreover, we will study the carcinogenic effects of CENPN by overexpressing it in normal liver cells, exploring its function in vivo, and ultimately develop a more effective strategy for the treatment of HCC patients.

### Funding
This work was supported by the National Natural Science Foundation of China (No. 81472799). The funders had no role in study design, data collection and analysis, decision to publish, or preparation of the manuscript.

### Grant Disclosures
The following grant information was disclosed by the authors:
National Natural Science Foundation of China: 81472799.

### Competing Interests
The authors declare there are no competing interests.

### Author Contributions
- Qingqing Wang and Yunfeng Zhou conceived and designed the experiments, authored or reviewed drafts of the paper, and approved the final draft.
- Xiaoyan Yu and Fengxia Chen performed the experiments, prepared figures and/or tables, and approved the final draft.
- Zhewen Zheng and Ningning Yang analyzed the data, prepared figures and/or tables, and approved the final draft.

### Data Availability
The main data are available at TCGA and NCBI-GEO (GSE87630, GSE112790 and GSE25097). The raw measurements are available in the Supplementary Files. https://portal.gdc.cancer.gov/.

### Supplemental Information
Supplemental information for this article can be found online at http://dx.doi.org/10.7717/peerj.11342#supplemental-information.

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
