# Peer review of "Centromere protein N may be a novel malignant prognostic biomarker for hepatocellular carcinoma"

_PeerJ, doi:10.7717/peerj.11342_

## Round 0.1 · original submission · Major Revisions

As you can see, opinions of the reviewers are ranging from minor revision to rejection. I decided to give you an opportunity to prepare revised version of the manuscript that would appropriately address all the critiques of the reviewers.

Reviewer 1 ·

Basic reporting

none

Experimental design

none

Validity of the findings

none

Additional comments

The 2 GEO Datasets used in this study is already analyzed and published in other reputed journals. Hence this is a reanalyzed study and no new investigations are carried out. The author mentioned automated GEO2R tool to extract Up and Down regulated DEGs but limma R package to extract DEG's. The Key Experimental DEG Tables for each dataset is omitted in this manuscript, hence this work might be not a novel study. Heatmaps for each datasets are also omitted. Valcano plots are not clear and incomlete. GO and Pathway Enrichment analysis Plots are not clear and incomplete and it should be in Tabular form with gene, gene counts and statistical parameters. The PPI n/w construction is not clear and incomplete. Module analysis is also omitted. The Target Gene-mirna and Target Gene - TF n/w construction is omitted. ROC analysis validation of selected hub genes are omitted.

·

Basic reporting

I have carefully reviewed the manuscript entitled: " Centromere protein N may be a novel malignant prognostic biomarker for hepatocellular carcinoma" by Wang et al. In this work, Wang et al. have identified CENPN as an important differentially expressed gene in hepatocellular carcinoma and the upregulation of CENPN was shown to be related with the low survival rate and advanced T classification of hepatocellular carcinoma patients. Moreover, in vitro study of cell proliferation and the damaging effect of X-rays on CENPN knocked down cells further justifies CENPN as a potential novel malignant prognostic biomarker for hepatocellular carcinoma in vitro. This paper is clearly written and well organized. The introduction and background are reasonable. Figures and tables are comprehensive and helpful. I have few minor concerns that could be addressed to improve the quality of this study.

1. Is there any specific reason behind choosing HepG2 and Huh7 cell lines among all the available hepatocellular carcinoma cell lines to study the oncogenic role of CENPN?

2. Was the effect of CENPN on major EMT markers and transcription factors checked in hepatocellular carcinoma cells? It would be worth checking the effect of CENPN knockdown on the levels of EMT markers in HCC cells to understand the oncogenic potential of CENPN.

Experimental design

no comment

Validity of the findings

no comment

Reviewer 3 ·

Basic reporting

In this study, Wang et al. proposed a bioinformatics analysis to understand the role of centromere protein N in hepatocellular carcinoma (HCC). They also validated the results by in vitro experiments and confirmed that this protein might be a novel prognostic biomarker of HCC. In my opinion, the study is well-prepared and holds the potential for publication. I raise some major comments to improve the study as follows:

English language should be improved. Currently, there are still some grammatical errors and typos such as:
- There is an urgent to discover ...
- ... liver cancer continues to be one of the leading 10 causes for cancer-related fatalities ...
- The level of expression of the hub genes in clinical samples were compared with those in - ...

An important concern of this study is that the authors missed discussing a lot of literature review related to bioinformatics analysis on HCC.

Experimental design

It needs to provide more information for methodology. Source codes should be provided for replicating the methods and results.

The authors selected different microarray data from GEO as well as TCGA. Did they concern about the batch effect removal among different data?

There must have space before the reference number.

GO database or analysis has been used in previous studies such as PMID: 31277574 and PMID: 31921391. Therefore, it is suggested to provide more references in this description.

Validity of the findings

The findings are good. The authors also validated the predictive performance using in vitro experiments.

Additional comments

No comment

---

## Round 0.2 · accepted · Accept

All critiques were completely addressed and the revised manuscript is acceptable now.

·

Basic reporting

In this study, Wang et al. have done bioinformatics analysis and in vitro experiments to study centromere protein N's role in hepatocellular carcinoma (HCC). The revised manuscript is well written, and the authors have added more updated references. Moreover, the authors have well addressed the reviewers’ concerns.

Experimental design

no comment

Validity of the findings

no comment

Reviewer 3 ·

Basic reporting

No comment.

Experimental design

No comment.

Validity of the findings

No comment.

Additional comments

No comment.